# The Key Physiological and Biochemical Traits Underlying Common Bean (*Phaseolus vulgaris* L.) Response to Iron Deficiency, and Related Interrelationships

Khawla Nsiri [1,2] and Abdelmajid Krouma [1,2,*]

[1] Laboratory of Ecosystems and Biodiversity in Arid Land, Faculty of Sciences, University of Sfax, Sfax 3000, Tunisia; khawlansiri206@gmail.com

[2] Faculty of Sciences and Techniques, University of Kairouan, Sidi Bouzid 9100, Tunisia

* Correspondence: abdelmajid.krouma@fstsbz.u-kairouan.tn; Tel.: +21-69-8580-069

**Abstract:** Iron deficiency is a worldwide nutritional problem affecting crop production. In Tunisia, this mineral disorder hampers the growth and yield of the major crops due to the abundance of calcareous soils that limit iron availability. The common bean is one of these crops suffering from lime-induced iron chlorosis. The exploration of the variability of common bean responses to iron deficiency allows us to screen tolerant cultivars and identify useful traits and indicators for further screening programs. To this end, two common bean cultivars (coco blanc, CB, and coco nain, CN) were cultivated hydroponically in standard nutrient solution (control) or nutrient solution deprived of iron (stressed). Analyses were reported on plant growth, photosynthetic pigments, photosynthesis, iron distribution, H-ATPase, and Fe-chelate reductase (Fe-CR) activities; important indicators were calculated; and convenient correlations were established. Current results demonstrated that iron deficiency stimulated specific symptoms of iron chlorosis on young leaves that were more precocious and severe in CB than CN. Spad index and chlorophyll pigments measurement confirmed these morphological changes and cultivar differences. Net photosynthesis (Pn) showed the same scheme of variation, with a significant decrease in Pn while respecting the previous cultivar's variability. Plant growth is no exception to this general trend. The biomass decrease was two times higher in CB than CN. Otherwise, this mineral disorder significantly decreased Fe concentration in all plant organs. However, CN accumulated 40% more Fe than CB, resulting from its higher Fe Fe-CR and H-ATPase activities. Our results also demonstrated the close dependence of these metabolic functions on Fe availability in shoots and the strict relationship between Fe-CR and H-ATPase, photosynthesis, and chlorophyll content. Otherwise, CN demonstrated higher efficiency of Fe's use (FeUE) for the key metabolic functions (photosynthesis, chlorophyll biosynthesis, and plant growth). The relative tolerance of CN as compared to CB was explained by its ability to establish a functional system less vulnerable to iron deficiency that operates effectively under problematic conditions. This system involves metabolic functions in shoots (photosynthesis, chlorophyll biosynthesis, Fe repartition, etc.) and others in roots (H-ATPase, Fe-CR), which are strictly interdependent.

**Keywords:** Fe chelate reductase; Fe use efficiency; photosynthetic pigments; photosynthesis; rhizosphere acidification





## 1. Introduction

Iron (Fe) is one of the essential elements for plant growth and reproduction. Being highly abundant on the planet, Fe is a micronutrient and is required by plants in small amounts. The most abundant form of Fe in soils is ferric oxide ($Fe_2O_3$) or hematite, which is extremely insoluble and imparts a red color to the soil. In aerated plant production agrosystems (pH $\pm$ 6.0), the concentrations of ferric ($Fe^{3+}$) and ferrous ($Fe^{2+}$) iron are very low (around $10^{-15}$ M), and the oxide, hydroxide, and phosphate forms control the concentration of Fe in the soil solution and its availability to plants. As pH increases by

one unit, the activity of $Fe^{3+}$ decreases 1000 times due to the formation of insoluble $Fe^{3+}$ hydroxide [1].

In plant metabolism, Fe is involved in many important compounds and physiological processes. Iron is required for the manufacturing process of chlorophyll biosynthesis and enzyme functions [2]. The involvement of Fe in chlorophyll synthesis is the reason for the chlorosis (yellowing) associated with Fe deficiency [3]. Iron is found in iron-containing heme proteins, such as cytochromes commonly observed in the electron transfer chains of the chloroplasts and mitochondria [4]. Iron is also associated with certain non-heme proteins, such as ferredoxin [5].

Iron uptake by plants is not as simple as with other essential elements. Iron is taken up by plant roots mainly in the cell elongation and maturation zones, about 1 to 4 cm behind the root tip. The uptake of Fe by the plant is an active process that is energy-dependent [6]. However, iron uptake is dependent on the plant's ability to reduce $Fe^{3+}$ to $Fe^{2+}$ and remove it from the complex or chelating compound. Previous studies have demonstrated that this reduction occurs at the root cell surface in the area of 1 to 4 cm behind the root tip [7], where the most protons and reductants are also released. At the root surface, Fe is generally reduced and removed from the chelating molecule, then diffused across the cell membrane. Otherwise, Fe uptake can be interfered with by other cations in the soil solution, such as manganese (Mn) and calcium (Ca).

In the plant, Fe exists mainly in the ferric form, and much of the Fe is found in the plastids. Another significant pool of Fe is found in the apoplast (extracellular area) of the older root parts, but it is thought this pool plays little role in supplying shoots with Fe at the installation of Fe deficiency [8]. Iron is relatively immobile once incorporated into compounds in the upper parts of plants. Re-translocation of Fe from one plant tissue to another is negligible [9]. Therefore, Fe deficiency symptoms appear first in the youngest leaves [10]. These juvenile organs show yellowing, sometimes referred to as "iron chlorosis". Soils with an alkaline pH (pH > 7.2) often result in Fe deficiency because Fe rapidly turns up in the oxide and unavailable form [11]. Acidic soils can also lead to Fe deficiency, likely due to manganese competition with Fe uptake [12]. High Ca and carbonate concentrations in soil from over-liming can also lead to Fe deficiencies ("lime-induced Fe chlorosis") due to the high pH and competition with Ca at the root surface [13]. In fact, over the past few decades, Fe deficiency responses have been intensively studied in Arabidopsis [14], common bean [15], soybean [16], rice [17], apple [18], tomato [19], and many other species. Plants respond with morphological and physiological modifications at the root level, involving complex local and systemic signaling machinery [20].

When subjected to Fe deficiency, plants adopt two main mechanisms for Fe acquisition from the soil solution. The first one, assigned as strategy I, is adopted by dicots and non-graminaceous monocots and uses a reduction mechanism. Strategy I plants develop a series of physiological responses at the root surface in order to remobilize insoluble Fe in the soil solution. They stimulate rhizosphere acidification (mediated by membrane enzymes such as $H^+$-ATPases), accumulate organic acids, and/or secrete chelating and reducing substances, mainly phenolic compounds, and induce the Fe-chelate reductase (Fe-CR) activity that reduces the insoluble $Fe^{3+}$ to the soluble form, $Fe^{2+}$. The Fe-CR activity is an obligatory step in iron uptake, and it is the main factor that makes iron available for uptake by all plants except grasses. It has been suggested that Fe-CR is dependent on rhizosphere pH and plays a more general role in the regulation of cation uptake [10,15]. Numerous studies have shown that Fe-CR is very sensitive to pH and is inhibited in alkaline soils [13,21,22]. The strategy II species uses a chelation mechanism and occurs only in graminaceous plants where Fe is chelated to be uptaken by roots [23,24]. These plants are characterized by the extrusion of phytosiderophores, responsible for the mobilization of Fe from the rhizosphere [25].

The common bean (*Phaseolus vulgaris* L.) uses the mechanism I strategy to take up Fe from the soil solution, where a proton efflux acts to acidify the rhizosphere via the action of $H^+$-ATPase [26], allowing ferric reductase oxidase to reduce insoluble $Fe^{3+}$ into

absorbable $Fe^{2+}$ [27]. Then, $Fe^{2+}$ is subsequently imported from the soil into the root cell through an iron-regulated transporter [28,29]. The present study focuses on this species, which belongs to the Fabaceae family. We intended to adopt an analytical dissection of some physio-biochemical traits of common bean subjected to Fe deficiency, implying metabolic changes in shoots and roots, in order to identify the common threat underlying its general behavior in response to this mineral constraint and the subsequent genotypic differences. This approach allows for understanding the interrelationships between the key metabolic functions in the different plant organs, highlighting the functional traits underlying common bean tolerance to iron deficiency, and identifying useful traits of tolerance. Photosynthesis and photosynthetic pigments, H-ATPase, and Fe-CR were deeply analyzed in relation to Fe uptake and compartmentation. Two cultivars were used: coco blanc with white seeds and coco nain with colored seeds.

## 2. Materials and Methods

### 2.1. Experimental Design and Growth Conditions

A greenhouse experiment was conducted on two common bean cultivars: coco blanc with white seeds and coco nain with colored seeds. Homogeneous seeds were soaked in 1 Kg pots equipped with a flow solution recovery plate and containing inert, clean quartz sand washed with concentrated sulfuric acid and rinsed several times with distilled water [30]. The experiment was conducted in the Faculty of Sciences and Techniques of Sidi Bouzid ($35°2'7.58''$ N, $9°29'2.18''$ E), under natural light and a temperature of 25 °C/ 17 °C ($\pm$2 °C, day/night), relative humidity about 75%, and using the following nutrient solutions [31]: $Ca(NO_3)_2$ (1.25 mM), $KNO_3$ (1.25 mM), $KH_2PO_4$ (1.60 mM), $MgSO_4$ (0.50 mM), $K_2SO_4$ (1.50 mM), $CaSO_4$ (3.50 mM), $H_3BO_3$ (4 µM), $MnSO_4$ (4 µM), $ZnSO_4$ (1 µM), $CuSO_4$ (1 µM), $CoCl_2$ (0.12 µM), $(Na)_6(Mo)_7O_{24}$ (0.12 µM). The excess solution recovered in the plate was put back into the pots the next day. At the emergence of young seedlings (7 days after germination), plants were separated into two plots: the first one (10 pots containing 10 plants) received the above nutrient solution added with 45 µM Fe as K-Fe-EDTA (control plants), and the second plot (10 pots containing 10 plants) received the same nutrient solution without iron (stressed plants). The nutrient solution was renewed weekly. The general behavior of plants was monitored daily to control the apparition of chlorosis symptoms that will be illustrated by photographs.

At the end of the experiment (28 DAG) and after nondestructive measurements (spad index, photosynthesis and gas exchange, $H^+$-ATPase, Fe-CR), plants were harvested, separated into shoots and roots, dried at 60 °C for 72 h, and then pulverized into a fine powder for further analysis. When harvested, roots were soaked in a 0.01 M $CaCl_2$ solution and washed thoroughly and successively in 3 baths of ultra-pure water to avoid contamination with elements from the soil [32].

### 2.2. Spad Index

Measurements on relative chlorophyll content were assessed using a SPAD-502 (Konica Minolta, Tokyo, Japan) before the gas exchange measurements on the second fully expanded leaf. Ten plants for each cultivar and treatment (control and stressed) were used. Values are expressed in SPAD units.

### 2.3. Net Photosynthesis and Gas Exchange

In order to homogenize measurements and obtain significant results, gas exchange parameters were measured on the same second fully expanded leaf (of the same plants used for Spad) using a portable photosynthesis system (CI-340, Camas, WA, USA). The induction of photosynthesis was ensured by photon flux of 1000 µmol m$^{-2}$ s$^{-1}$. The other parameters were maintained constant: sample $pCO_2$ at 362 mbar, flow rate at 500 µmol$^{-1}$, and temperature at 25 °C [33].

### 2.4. Acidification Capacity (H-ATPase Activity)

Rhizosphere acidification capacity (RAC) was measured before the final harvest on intact plants removed from their pots. Roots of ten plants were transferred individually into 200 mL of a continuously aerated solution containing 10 mM KCl + 1 mM $CaCl_2$, adjusted to pH 6.0 with 0.1 N NaOH [34]. pH values were measured every 30 min.

### 2.5. Fe Chelates Reductase Activity (Fe-CR)

Fe-CR activity was measured the next day after H-ATPase measurements on the same plants according to the method of Krouma et al. [34]. Roots were soaked in 50 mL of a solution containing 0.5 mM $CaSO_4$, 50 μM Fe(III)EDTA, 50 mM Tris, and 0.3 mM Ferrozine, adjusted to pH 7.5 with $H_2SO_4$. The absorbance of the Fe(II) complex was measured at 535 nm based on the extinction coefficient of 22 mM $Cm^{-1}$.

### 2.6. Determination of Photosynthetic Pigments

The photosynthetic pigments were determined according to the method of Arnon [35]. The extraction of chlorophyll was performed on 50 mg of fresh leaf material in 10 mL of 80% acetone. After 48 h in the dark, the extract was collected, and the absorbance was then read at 663 and 645 nm, and chlorophyll a (chl-a), chlorophyll b (chl-b), and total chylorophyll (chl-tot) concentrations were calculated.

### 2.7. Determination of Active Iron

To analyze the fraction of active iron in the different plant organs, 25 mg of finely powdered dry matter is transferred to continuously agitated tubes containing 5 mL of 1N hydrochloric acid solution for 4 h. The extraction was stopped by adding 5 mL of deionized water, and the extract was filtered. Fe content was determined by the atomic absorption spectrophotometry method according to Köseoglu and Açikgöz [36].

### 2.8. Calculation and Analyzed Parameters

$$\text{Chlorophyll a: Chl-a} = (12.7 \times A_{663}) - (2.69 \times A_{645})$$

$$\text{Chlorophyll b: Chl-b} = (22.9 \times A_{645}) - (4.68 \times A_{663})$$

$$\text{Total chlorophyll: Chl-tot} = (20.2 \times A_{645}) + (8.02 \times A_{663})$$

$$\text{Fe quantity in shoots: FeQ-Sh} = \text{FeSh} \times \text{DWSh}$$

where FeSh represents the concentration of Fe in shoots, DWSh represents the dry weight biomass produced by shoots.

$$\text{Fe quantity in roots: FeQ-R} = \text{FeR} \times \text{DWR}$$

where FeR represents the concentration of Fe in roots, DWR represents the dry weight biomass produced by roots.

$$\text{Total Fe quantity: FeQ-tot} = \text{FeQ-Sh} + \text{FeQ-R}$$

$$\text{Fe translocation; FeT} = \text{FeQ-Sh}/\text{FeQ-tot}$$

where FeQ-Sh represents the quantity of Fe calculated in shoots, FeQ-tot represents the quantity of total Fe calculated in the plant.

$$\text{Fe use efficiency for chlorophyll biosynthesis: FeUE-chl} = \text{Chl-tot}/\text{FeSh}$$

where Chl-tot represents the concentration of total chglorophyll, FeSh represents the concentration of Fe in shoots.

$$\text{Fe use efficiency for plant growth: FeUE-DW} = \text{DW/FeSh}$$

where DW represents the dry weight biomass produced by the plant, FeSh represents the concentration of Fe in shoots.

$$\text{Fe use efficiency for photosynthesis: FeUE-Pn} = \text{Pn/FeSh}$$

where Pn represents the net photosynthesis, FeSh represents the concentration of Fe in shoots.

$$\text{Membrane efficiency: ME-E} = \text{Fe-CR/pH}$$

where Fe-CR represents the Fe chelate reductase, pH represents the pH measured in the rhizosphere solution.

$$\text{Rhizosphere acidification capacity: RAC} = \text{pHs} - \text{pHc}$$

where pHs represents the pH in the solution of stressed plants, pHc represents the pH in the solution of control plants.

$$\text{Fe chelates reductase capacity: FeCRC} = (\text{Fe-CRs}) - (\text{Fe-CRc})$$

where Fe-CRs represents the Fe-CR in stressed plants, Fe-CRc represents the Fe-CR in control plants.

*2.9. Statistical Analysis*

Data and statistical analyses were performed using the software StatPlus Pro, 2023. All data were presented as mean ± standard error. Analysis of variance (ANOVA) was performed to check whether the effects of Fe treatment (C and S) on the respective factor were significant. The significance of differences among treatments was determined by Fisher's least significant difference test (LSD) at 5%. Means were declared significantly different when the difference between any two treatments was more important than the LSD value generated from the ANOVA. They were marked by different letters in the figures and tables.

**3. Results**

*3.1. Photosynthetic Pigments, Gas Exchange Parameters, and Plant Growth*

Before the necessary analyses, the plants were subject to daily monitoring of their morphological behavior. Thus, we noticed the appearance of specific symptoms of iron deficiency, commonly known as Fe-chlorosis, which affect specifically the young leaves of the plants subjected to iron deficiency. These symptoms appeared in coco blanc after 15 days of treatment and after 17 days in coco nain. It should be noted that the chlorosis that appeared in coco blanc was severe and evolved towards necrosis. In coco nain, the chlorosis remained moderate until the end of the experiment.

Otherwise, the measured spad index confirmed the observed changes. Table 1 showed that subjecting plants to Fe deficiency significantly decreased the spad index in all plants, with some differences. In CB, spad decreased by 46% as compared to control plants, whereas in CN, this decrease remained significant but did not exceed 21%. These cultivar differences appeared under limited Fe nutrition, where CN that showed moderate chlorosis expressed a spad index 1.6 times higher than CB. In fact, shoot chlorosis usually reflects disturbances in chlorophyll biosynthesis. To deeply analyze this phenomenon, we analyzed the photosynthetic pigments. Table 1 showed that F deficiency significantly decreased chl-a, chl-b, and chl-tot in the two studied cultivars and confirmed the previously observed differences. As compared to control plants, chl-a decreased by 65% and 83%, chl-b decreased

by 68% and 84%, and chl-tot decreased by 66% and 86%, respectively, in CN and CB. Indeed, all chlorophyll pigments were affected in the same way, with respect to cultivar differences. While remaining less affected, CN accumulates 1.8 times chl-a, 1.4 times chl-b, and 2.2 times chl-tot as compared to CB.

**Table 1.** Chlorophyll a (chl-a), chlorophyll b (chl-b), total chlorophyll (chl-tot), spad index, and plant dry weight biomass (DW) in common bean plants cultivated on adequate nutrient solution (C) or subjected to iron deficiency (S). Within rows, means with the same letter are not significantly different at $\alpha$ = 0.05 according to Fisher's least significant difference. Standard error of the mean (n = 10).

| | CN | | CB | |
|---|---|---|---|---|
| | Control | Stressed | Control | Stressed |
| Chl-a (mg g$^{-1}$ FW) | 510 ± 32 [b] | 179 ± 15.2 [c] | 574 ± 41.1 [a] | 99 ± 8.4 [d] |
| Chl-b (mg g$^{-1}$ FW) | 233 ± 18.5 [b] | 74 ± 6.3 [c] | 342 ± 30.4 [a] | 54 ± 4.2 [d] |
| Chl-tot (mg g$^{-1}$ FW) | 742 ± 33.2 [b] | 253 ± 18.8 [c] | 815 ± 51.2 [a] | 116 ± 8.9 [d] |
| spad index | 34.57 ± 3.3 [a] | 27.31 ± 2.5 [b] | 31.81 ± 2.5 [ab] | 17.19 ± 1.5 [c] |
| DW (g plant$^{-1}$) | 683 ± 53.3 [a] | 543 ± 32.4 [c] | 581 ± 41.5 [b] | 334 ± 22.6 [d] |

The evaluation of the effect of iron deficiency on plant growth shows that plants adequately supplied with iron (control) do not show significant cultivar differences. However, this nutritional constraint significantly decreased plant growth. Table 1 shows that biomass production decreased by 43% in CB and 20% in CN under iron deficiency as compared to control plants. The superiority of CN under limited Fe nutrition was also maintained at this level. The biomass produced by CN exceeded that of CB by 63% in this problematic condition.

Measurements made on net photosynthesis (Pn) demonstrated that Fe deficiency significantly affects this key metabolic function in the common bean. In fact, Figure 1a showed that Pn decreased by 65% in CN and by 78% in CB as compared to control plants. As previously shown, CN maintains better photosynthetic activity under limited Fe nutrition (+65%) as compared to CN. The other gas exchange parameters, evapotranspiration (Figure 1b) and stomatal conductance (Figure 1c), even though they decreased under iron deficiency, remained, in a way, less affected than Pn.

### 3.2. Iron Nutrition and Distribution

The physiologically active fraction of iron (Fe$^{2+}$) was analyzed in the plant organs (Figure 2). With respect to cultivar variability, Fe concentration significantly decreased in shoots (a) as well as roots (b) of common bean plants subjected to Fe deficiency. As compared to control plants, this decrease reached $-33$ and $-46$% in shoots and $-35$% and $-45$% in roots, respectively, in CN and CB subjected to iron deficiency. Otherwise, these concentrations remained higher in all plant organs of CN as compared to CB.

To deeply analyze Fe nutrition and distribution in common bean cultivars subjected to iron chlorosis, we calculated Fe quantities accumulated in plant organs and the whole plant. Table 2 demonstrated that iron starvation significantly hampered Fe accumulation in shoots and roots, then in the whole plant, with some cultivar differences. Fe quantities decreased by 53% and 42% in shoots (FeQ-sh), by 84% and 58% in roots (FeQ-R), and by 66% and 59% in whole plants (FeQ-tot), respectively, in CB and CN subjected to Fe deficiency as compared to control plants (Table 2). Thus, the previously observed differences were also maintained at this level, with a special superiority for CN. In addition to these data, we calculated Fe translocation (FeT, Table 2). This trait illustrated the fraction of Fe allocated to shoots compared to the total quantities of Fe in the plant. Table 2 shows that FeT increased in plants subjected to iron deficiency, reflecting the preferential allocation of this micronutrient to shoots under problematic conditions. The cultivar that demonstrated better behavior regarding all previously presented traits (CN) maintained its superiority, with shoot Fe representing 52% of the total plant Fe, against 48% in CB.

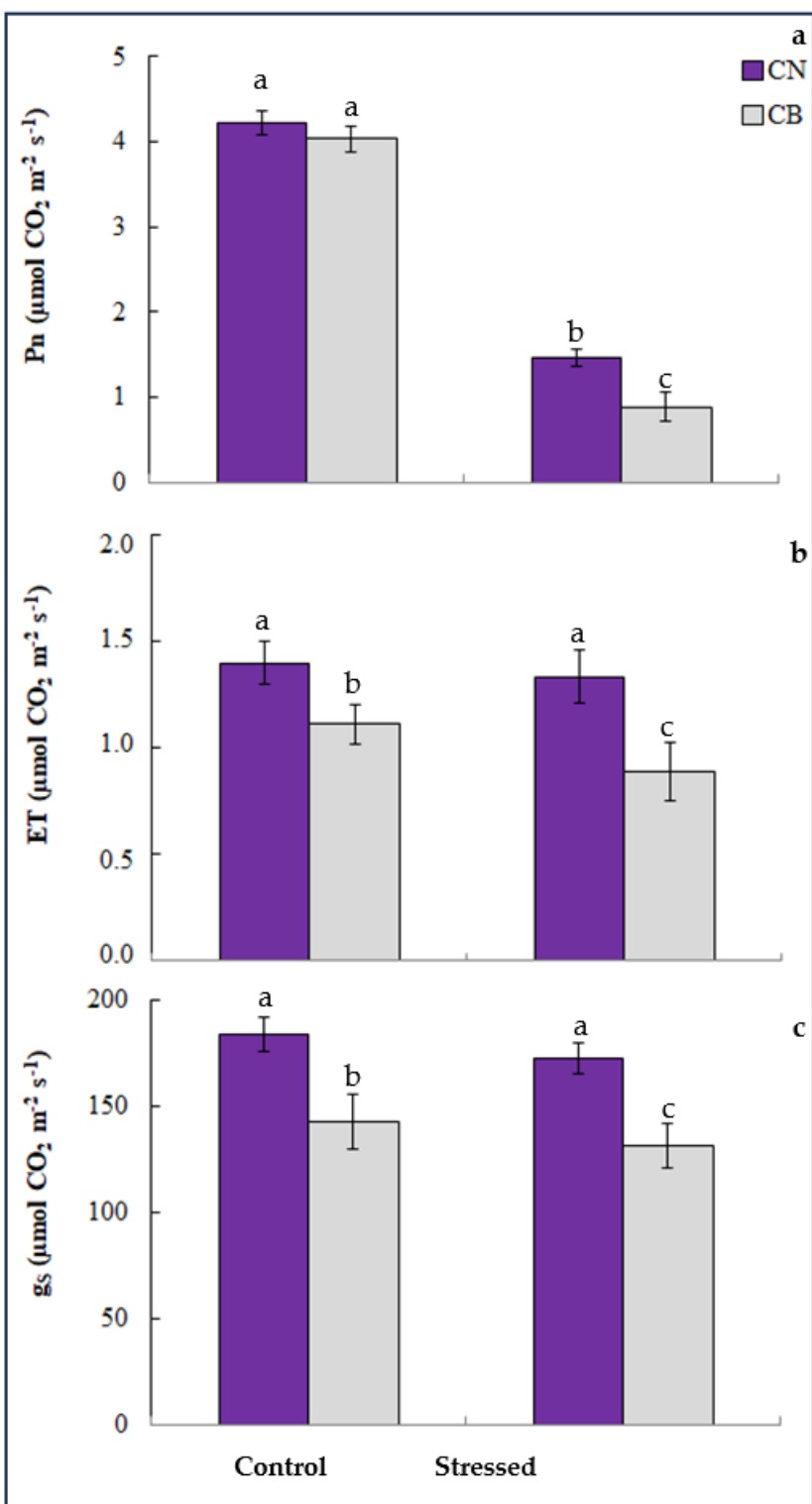

**Figure 1.** Net photosynthesis (Pn, (**a**)), evapotranspiration (ET, (**b**)), and stomatal conductance (gs, (**c**)) in common bean plants cultivated on adequate nutrient solution (Control) or subjected to iron deficiency (Stressed). Within histogram, means with the same letter are not significantly different at $\alpha = 0.05$ according to Fisher's least significant difference. Bars on the columns represent the standard error of the mean (n = 10).

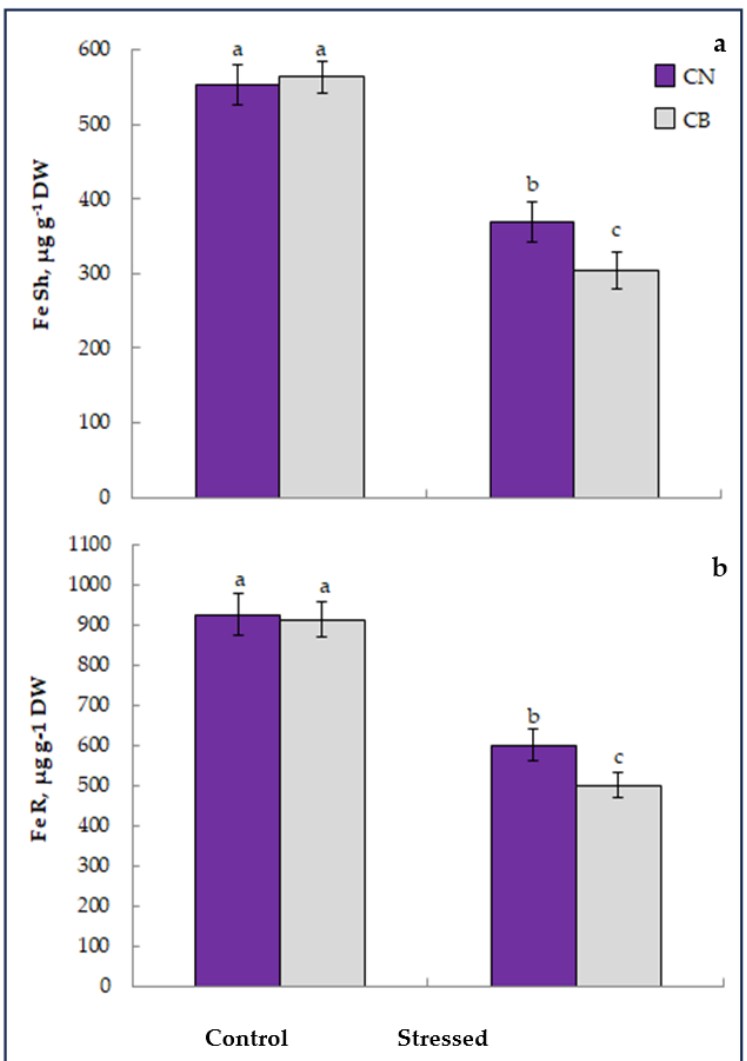

**Figure 2.** Active Fe ($Fe^{2+}$) concentration in shoots (**a**) and roots (**b**) of common bean plants adequately supplied with iron (Control) or subjected to iron deficiency (Stressed). Within histogram, means with the same letter are not significantly different at $\alpha = 0.05$ according to Fisher's least significant difference. Bars on the columns represent the standard error of the mean (n = 10).

**Table 2.** Quantities of Fe ($\mu g \ plant^{-1}$) in shoots (FeQ-Sh), roots (FeQ-R), and in the plant (FeQ-tot), and Fe translocation (FeT) in common bean plants cultivated on adequate nutrient solution (Control) or subjected to iron deficiency (Stressed). Within rows, means with the same letter are not significantly different at $\alpha = 0.05$ according to Fisher's least significant difference. Standard error of the mean (n = 10).

| | CN | | CB | |
|---|---|---|---|---|
| | Control | Stressed | Control | Stressed |
| FeQ-Sh | 1505 ± 48.5 [a] | 873 ± 36.6 [c] | 1375 ± 58.5 [b] | 642 ± 33.7 [d] |
| FeQ-R | 2624 ± 101.1 [a] | 807 ± 43.4 [b] | 2559 ± 121.3 [ab] | 686 ± 44.6 [c] |
| FeQ-tot | 4128 ± 125.3 [a] | 168 ± 88.6 [c] | 3933 ± 142.4 [b] | 1328 ± 78.4 [d] |
| FeT (%) | 36 ± 2.7 [b] | 52 ± 4.5 [a] | 35 ± 2.9 [b] | 48 ± 3.8 [a] |

### 3.3. Rhizosphere Acidification (H-ATPase) and Fe-Chelate Reductase (Fe-CR) Activities

Measurements made on rhizosphere acidification in control plants showed unstable fluctuations above the initial pH (6.0) without reaching the state of rhizosphere acidification. However, plants suffering from iron deficiency significantly decreased their rhizosphere pH,

which increased with time, reaching its maximum after 120 mn (Figure 3a). As compared to CB, the cultivar CN expressed a higher capacity for pH decrease, reaching the lowest value of 4.88, compared to 5.7 for CB. The calculation of rhizosphere acidification capacity (RAC), expressed as the difference between pH control and pH stress (Table 3), showed an increasing RAC with time until 120 mn, then decreasing. However, RAC in CN exceeded that in CB, independently of time. For example, at its maximum activity (120 mn), RAC in CN was 1.9 times higher than that of CB.

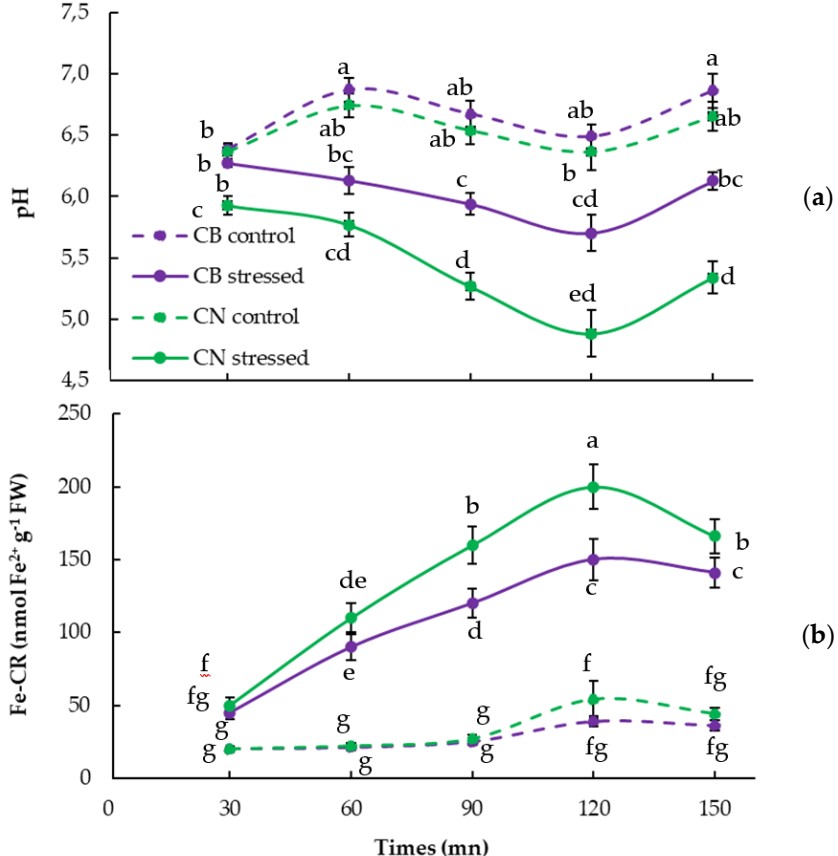

**Figure 3.** Rhizosphere acidification (**a**) and Fe chelates reductase activity (Fe-CR, (**b**)) in common bean plants subjected (stressed) or not (control) to iron deficiency. Plants were incubated in specific solutions (see MM) and pH and Fe-CR were recorded as a function of time. The letters in the figure indicate the significance of differences. According to Fisher's least significant difference, means with the same letter are not significantly different at $\alpha$ = 0.05. Bars represent the standard error of the mean (n = 10).

**Table 3.** Rhizosphere acidification capacity (RAC) and Fe chelates reductase capacity (Fe-CRC) in common bean plants cultivated on adequate nutrient solution (Control) or subjected to iron deficiency (Stressed). Within rows, means with the same letter are not significantly different at $\alpha$ = 0.05 according to Fisher's least significant difference. Standard error of the mean (n = 10).

| Time (mn) | RAC | | Fe-CRC | |
|---|---|---|---|---|
| | CB | CN | CB | CN |
| 30 | $-0.11 \pm 0.01$ [f] | $-0.44 \pm 0.04$ [e] | $25 \pm 2.2$ [h] | $30 \pm 2.6$ [g] |
| 60 | $-0.74 \pm 0.06$ [de] | $-0.97 \pm 0.07$ [c] | $69 \pm 5.3$ [f] | $88 \pm 6.4$ [ef] |
| 90 | $-0.73 \pm 0.05$ [de] | $-1.27 \pm 0.12$ [bc] | $95 \pm 6.9$ [e] | $133 \pm 10.2$ [b] |
| 120 | $-0.79 \pm 0.06$ [d] | $-1.48 \pm 0.14$ [a] | $111 \pm 8.4$ [d] | $146 \pm 10.5$ [a] |
| 150 | $-0.74 \pm 0.05$ [de] | $-1.31 \pm 0.11$ [b] | $105 \pm 9.0$ [de] | $122 \pm 9.7$ [c] |

Otherwise, we measured the Fe-chelate reductase activity in the same plants used for rhizosphere acidification. Figure 3b showed no significant modifications in the control plants. However, plants suffering from iron chlorosis demonstrated a significant increase in Fe-CR with time, reaching their maximum after 120 mn, beyond which Fe-CR decreased. The cultivar CN maintained its superiority as compared to CB. The calculation of Fe chelates reductase capacity (Fe-CRC, Table 3) expressed as the difference between Fe-CR control and Fe-CR stressed demonstrated higher values in CN than CB, independently of incubation time. The maximum of Fe-CRC was reached after 120 mn.

To progress in the elucidation of the tolerance indicators, we calculated the membrane efficiency (Mb-E), expressed as the quantity of Fe reduced by pH unit (Table 4). Obtained results demonstrated that Mb-E significantly increased in stressed plants as compared to control ones (Mb-E increased 2 to 5 times in CB and 2 to 7 times in CN, depending on incubation time). The cultivar differences were also maintained at this level; Mb-E was usually higher in CN than CB in stressed plants. At maximum AC and Fe-CRC (120 mn), Mb-E was 1.6 times higher in CN as compared to CB subjected to iron deficiency.

**Table 4.** Membrane efficiency (Mb-E) in root common bean plants cultivated on standard nutrient solution (Control), or Fe-deficient solution (Stressed). Within rows, means with the same letter are not significantly different at $\alpha = 0.05$ according to Fisher's least significant difference. Standard error of the mean (n = 10).

| Mb-E | CB | | CN | |
|---|---|---|---|---|
| Time (mn) | Control | Stressed | Control | Stressed |
| 30 | $3.1 \pm 0.33$ [l] | $7.2 \pm 0.58$ [h] | $3.1 \pm 0.30$ [l] | $8.4 \pm 0.62$ [g] |
| 60 | $3.1 \pm 0.25$ [l] | $14.7 \pm 1.44$ [f] | $3.3 \pm 0.27$ [l] | $19.1 \pm 1.7$ [e] |
| 90 | $3.7 \pm 0.33$ [kl] | $20.2 \pm 1.8$ [e] | $4.1 \pm 0.35$ [k] | $30.4 \pm 2.6$ [b] |
| 120 | $6.0 \pm 0.51$ [i] | $26.3 \pm 2.2$ [c] | $8.5 \pm 0.66$ [g] | $41.0 \pm 3.7$ [a] |
| 150 | $5.2 \pm 0.48$ [j] | $23.0 \pm 2.1$ [d] | $6.6 \pm 0.49$ [hi] | $31.1 \pm 2.9$ [b] |

*3.4. Interrelationships and Physiological Indicators*

Regarding the diverse physiological and biochemical traits developed in this study (some of which operate in the rhizosphere while others in the shoots) and their esteemed relationships, we established several correlations between them. For the first time, we correlated the physiological key functions with shoot Fe. Figure 4 showed a positive and strict relationship between net photosynthesis and shoot Fe (a), between chlorophyll concentration and shoot Fe (b), and between plant growth and shoot Fe (c), testifying that these physiological functions are highly dependent on the available Fe in shoots.

The available Fe in shoots originates from the rhizosphere by direct uptake, if available, or Fe-CR activity under deficient conditions. In fact, Figure 5, which correlates the quantities of Fe accumulated in plants with Fe-CR, demonstrated a close negative relationship, illustrating that the increase in Fe-CR activity is still associated with a greater remobilization of Fe.

Otherwise, our results showed a strict dependence of Fe-CR on rhizosphere acidification. Figure 6, which correlates Fe-CR with rhizosphere pH, showed a strict negative relationship between these two functions, with a particular superiority for CN, which expressed better H-ATPase and Fe-CR activities.

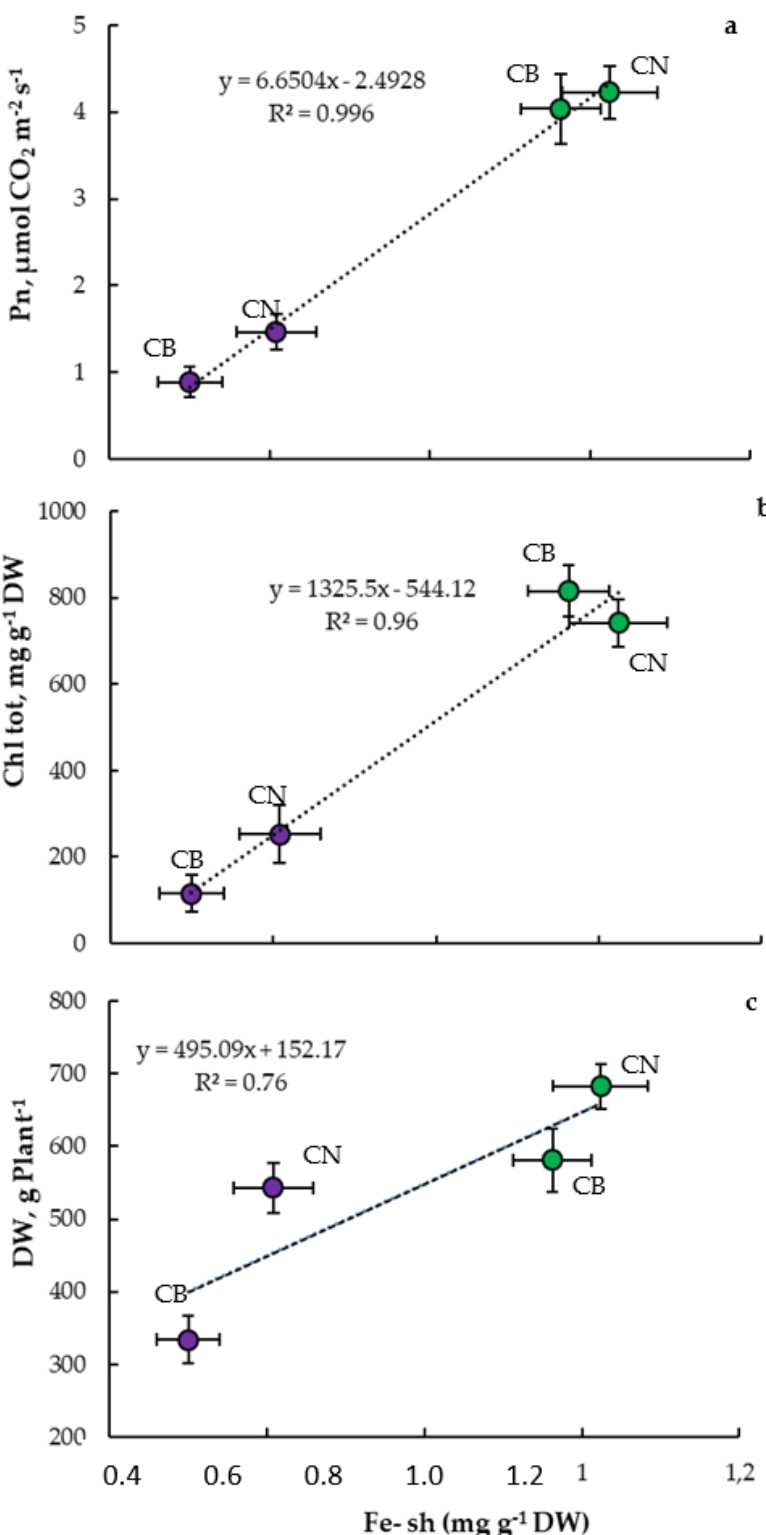

**Figure 4.** Relationship between photosynthesis and shoot Fe concentration (**a**), between chlorophyll content and shoot Fe concentration (**b**), and between plant growth and shoot concentration (**c**) in common bean plants subjected (Stressed) or not (Control) to iron deficiency. Vertical and horizontal bars represent the standard error of the mean (n = 10).

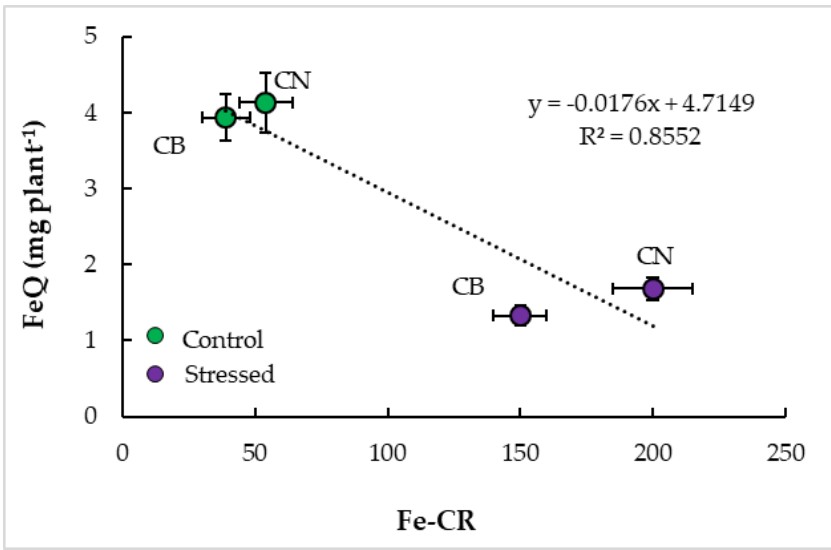

**Figure 5.** Relationship between plant Fe content and Fe-CR in common bean plants subjected (Stressed) or not (Control) to iron deficiency. Vertical and horizontal bars represent the standard error of the mean (n = 10).

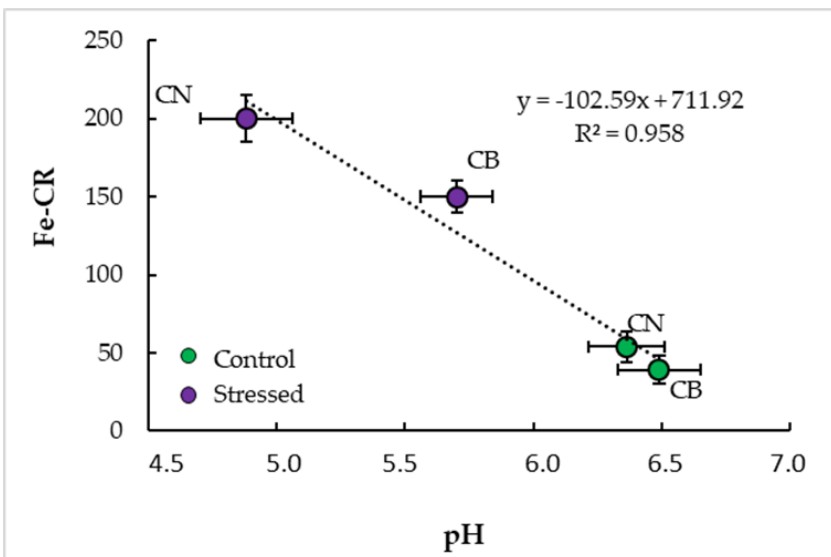

**Figure 6.** Relationship between Fe-CR and rhizosphere acidification in the roots of common bean plants adequately supplied with iron (Control), or subjected to iron deficiency (Stressed). Vertical and horizontal bars represent the standard error of the mean (n = 10).

Nevertheless, the photosynthetic activity, which represents the main source for biomass production, and the photosynthetic pigments, which ensure the necessary light harvesting for photosynthesis, are shown to be highly dependent (Figure 7). The superiority of CN was also expressed at this level, particularly under Fe-deficient conditions. As compared to CB, CN produced more chlorophyll and developed better photosynthetic activity.

In order to identify more indicators of Fe tolerance in the common bean, we calculated the Fe use efficiency for plant growth (FeUE-DW), the Fe use efficiency for photosynthesis (FeUE-Pn), and the Fe use efficiency for chlorophyll biosynthesis (FeUE-chl). Table 5 showed that FeUE-DW significantly increased in CN (+34%) and, with a lower significance, in CB (+10%), in plants subjected to Fe deficiency as compared to control ones. For FeUE-Pn, we noticed a significant decrease in stressed plants as compared to control ones (−42% in CN and −58% in CB), while FeUE-chl significantly decreased in plants suffering from Fe

deficiency (−43% in CN and −73% in CB). The main result emerging from this calculation is the clear superiority of CN under problematic conditions, which behaves more efficiently as compared to CB. FeUE-DW, FeUE-Pn, and FeUE-chl are 1.34, 1.36, and 1.79 times higher in CN as compared to CB, respectively.

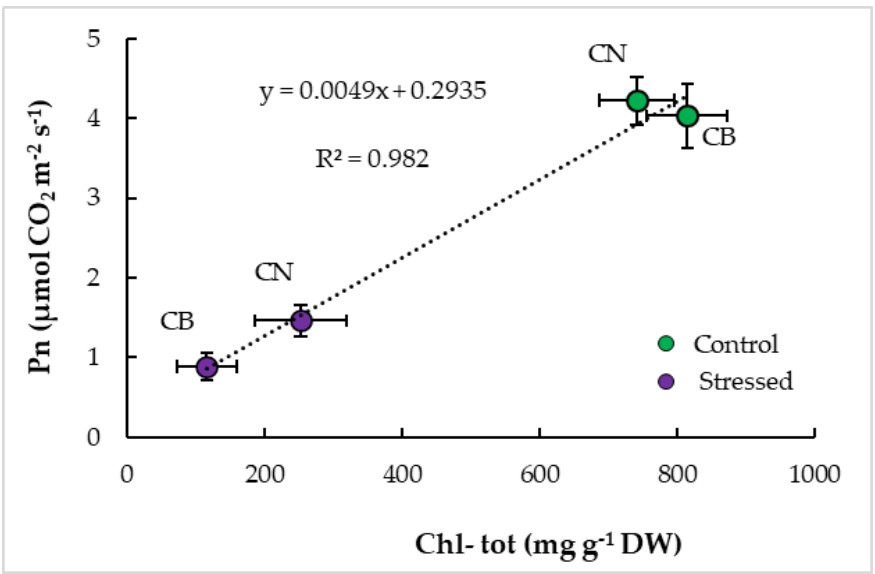

**Figure 7.** Relationship between net photosynthesis and chlorophyll content in common bean plants subjected (Stressed) or not (Control) to iron deficiency. Vertical and horizontal bars represent the standard error of the mean (n = 10).

**Table 5.** Fe use efficiency for plant growth (FeUE-DW), Fe use efficiency for photosynthesis (FeUE-Pn), and Fe use efficiency for chlorophyll biosynthesis (FeUE-chl) in common bean plants subjected (S) or not (C) to iron deficiency. Within rows, means with the same letter are not significantly different at $\alpha$ = 0.05 according to Fisher's least significant difference. Standard error of the mean (n = 10).

| | CN | | CB | |
|---|---|---|---|---|
| | Control | Stressed | Control | Stressed |
| FeUE-DW | 0.67 ± 0.063 [b] | 0.89 ± 0.072 [a] | 0.60 ± 0.054 [c] | 0.67 ± 0.057 [b] |
| FeUE-Pn | 4.13 ± 0.35 [a] | 2.41 ± 0.18 [b] | 4.19 ± 0.38 [a] | 1.77 ± 0.15 [c] |
| FeUE-chl | 0.72 ± 0.066 [b] | 0.42 ± 0.033 [c] | 0.85 ± 0.068 [a] | 0.23 ± 0.020 [d] |

Finally, we correlated the main physiological functions that determine the general behavior of plants (Pn and chl) with Fe use efficiency. Figure 8 showed a very strict positive relationship between chlorophyll accumulation and FeUE-ch. The same behavior was observed when correlating net photosynthesis with FeUE-Pn (Figure 9), with clear discrimination between cultivars. When cultivated under limiting Fe availability, CN showed higher FeUE-chl and FeUE-Pn with better chlorophyll biosynthesis and photosynthetic activity as compared to CB. This result allows us to suggest that, unlike their previously demonstrated performances (H-ATPase, Fe-CR, Pn, chl, etc.), CN used efficiently the minuscule available Fe.

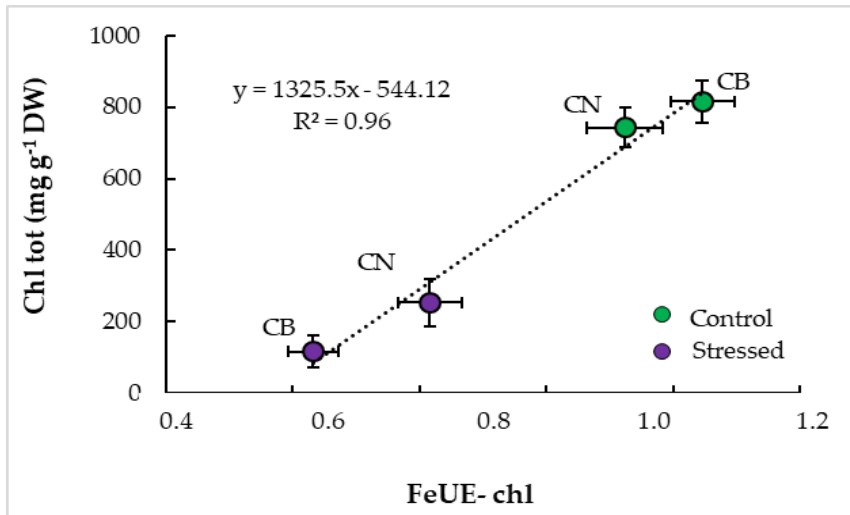

**Figure 8.** Relationship between chlorophyll content and FeUE-chl in common bean plants subjected (Stressed) or not (Control) to iron deficiency. Vertical and horizontal bars represent the standard error of the mean (n = 10).

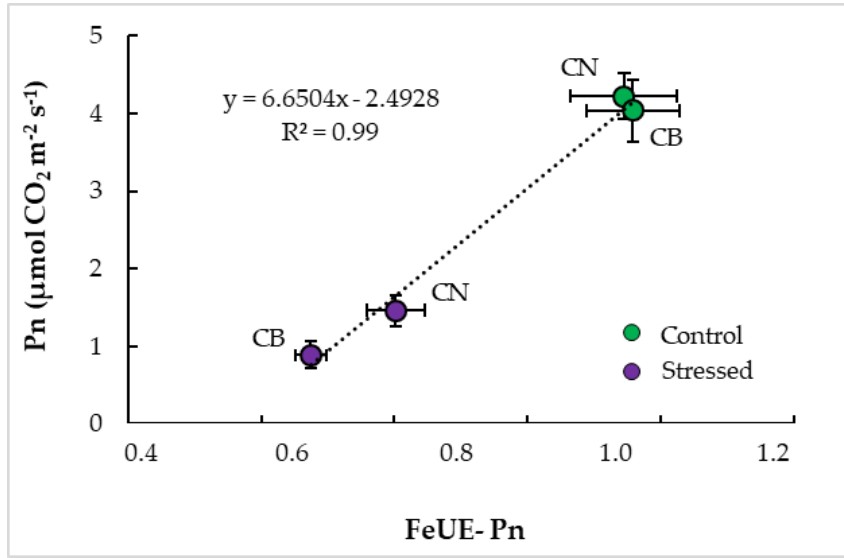

**Figure 9.** Relationship between net photosynthesis and FeUE-Pn in common bean plants cultivated on standard nutrient solution (Control), or Fe-deficient solution (Stressed). Vertical and horizontal bars represent the standard error of the mean (n = 10).

## 4. Discussion

Iron is of great importance for plant growth and health. This trace element plays an essential role in key metabolic processes such as DNA synthesis, energy transfer, respiration, and photosynthesis. Without iron, a plant is unable to produce chlorophyll and therefore stimulate photosynthesis and oxygen production. Since chlorophyll is responsible for the green color of the leaves, one of the symptoms of iron deficiency, or ferric chlorosis, is the development of yellow leaves with dark green veins. Other physiological, biochemical, and even molecular metabolic changes occur in plants suffering from iron deficiency and remain strictly dependent on cultivars, genotypes, and species.

### 4.1. Plant Aspect and Physiological Response to Iron Deficiency

The current study demonstrated that iron deficiency induced specific Fe chlorosis that was more severe and precocious in CB than CN. Consequently, chlorophyll concentrations

(chl-a, chl-b, and chl-tot) significantly decreased, photosynthesis was hampered, and plant growth was reduced. Iron uptake and distribution were also disturbed. Regarding all these traits, cultivar differences were observed, and CN showed higher plasticity to operate under this condition as compared to CB. These findings do not escape those previously obtained by other authors in diverse crop plants [10,37,38]. These authors identified specific symptoms of iron chlorosis whose onset and intensity depend on species and genotypes. In fact, Fe is known to be involved in chlorophyll and carotenoids biosynthesis [33,39,40], and any decrease in these pigments' concentration will be illustrated first by visual symptoms on young leaves. This is due to the fact that Fe has less mobility in the plant, and hence, its deficiency symptoms first appear in young leaves [41]. Otherwise, chlorophyll is a natural green pigment that plays a crucial role in photosynthesis [42–46], plastid protein metabolism [47], and plays a primordial role in key metabolic functions, such as electron transfer in photosynthesis and respiration [48,49]. Thus, Fe deficiency certainly impacts the plant's growth and development through these metabolic reactions. In light of this fact, the observed Fe-chlorosis and the decrease in photosynthetic pigments and photosynthesis in this study can be explained by a harmful reduction in the availability of Fe for these functions, resulting in a drop in growth. Moreover, the analyzed Fe concentration confirms this hypothesis by showing a drastic decrease in the shoots and roots of plants suffering from Fe deficiency. The restricted Fe translocation to shoots observed in this study (Table 2) can also explain the highlighted chlorosis caused by the low availability of Fe for chlorophyll biosynthesis. Accordingly, Bisht et al. [50] reported that iron deprivation decreases chlorophyll content in leaves and inhibits Fe translocation to growing plant tissues. Lopez-Millan et al. [51] confirmed these relationships and concluded that chlorophyll biosynthesis fully reflects the fraction of physiologically active iron available to the plant. Valipour et al. [3] revealed in quince seedlings a drastic decrease in chlorophyll, carotenoids, and Fe concentrations. In fact, it is well documented that the photosynthetic machinery benefits from the major Fe in cells, 80% in the chloroplast [52], and 21–22 atoms of Fe in the photosynthetic apparatus [53]. Other authors reported that iron deficiency decreased the electron transport capacity and reduced the yield of photosystems II and I, as well as the contents of cytochrome b559, cytochrome f, P700, and ferredoxin [54,55]. The $CO_2$ fixation in the Calvin cycle can also be inhibited through the restriction of some stromal enzymes under iron deficiency [53]. Zhao et al. [56] reported that, among these physio-biochemical disorders, iron deficiency can lead to a drastic reduction in yield and fruit quality.

### 4.2. Iron Remobilization and Use Efficiency

The overall obtained results argue in favor of CN tolerance as compared to CB. CN expressed lighter chlorosis symptoms, accumulated more chlorophyll, developed higher photosynthesis, and produced more biomass. At the same time, CN remobilized, uptake, and translocated more Fe to its shoots. Thus, we can suggest that the main reason for CN's tolerance to iron deficiency was its ability to improve Fe availability for all the above physiological and biochemical metabolic reactions. This is confirmed by the higher stimulation of Fe-CR in iron-deficient plants, where Fe-CRC reaches 1.3 times in CN as compared to CB at its maximum expression (120 mn). This better Fe-CR activity stimulation was also the result of a better capacity for rhizosphere acidification, where RAC reaches 1.9 times in CN as compared to CB at its maximum functioning (after 120 mn). These root-membrane functions were also the reason for the 27% more Fe accumulated in plants of CN as compared to plants of CB subjected to iron deficiency. Thus, a common thread that links the metabolic reactions in roots and those in shoots seems to exist and underlies the common bean tolerance to Fe deficiency. The cultivar that acidifies its rhizosphere, stimulates its Fe-CR, remobilizes more Fe, and then develops better metabolic functioning (chlorophyll biosynthesis, photosynthesis, and plant growth). However, the relative tolerance of CN cannot be explained only by rhizosphere acidification and Fe-CR activities. The calculated FeUE-chl, FeUE-Pn, and FeUE-DW revealed higher efficiency of Fe use as compared to CB.

The 79%, 36%, and 34% higher FeUE-chl, FeUE-Pn, and FeUE-DW, respectively, in CN as compared to CB subjected to iron deficiency can also be another reason for CN tolerance. This cultivar, other than its higher capacity of Fe remobilization under problematic Fe availability, used efficiently the minuscule available Fe for a better functioning of the Fe-dependent metabolic reactions.

### 4.3. Parameter Interrelationships and Tolerance Indicators

The established relationships between the key physio-biochemical functions (Figures 4–9) confirmed their interdependence and the existence of the above-mentioned common thread underlying CN tolerance. Chlorophyll, photosynthesis, and plant growth were highly dependent on the iron available in the shoots. Shoot Fe depends strictly on Fe-CR activity, which, in turn, is pH-dependent. Chlorophyll and photosynthesis are also interdependent, but both depend strictly on Fe use efficiency. Indeed, the literature review has taught us that the so-called strategy I plants, including the common bean, develop adaptive responses when subjected to iron deficiency that involve morphological, physiological, and biochemical changes starting in the roots as a first response [10,15]. These changes include, for the first time, rhizosphere acidification, chelating substances release, and stimulation of Fe-chelate reductase (Fe-CR) activity in roots; then the uptake of Fe was provided by a specific transporter. Thus, as demonstrated in this study, the enhanced rhizosphere acidification constitutes an adequate environment for Fe-CR activity that provides plants with the minimum need for Fe. This physio-biochemical mechanism operates more efficiently in CN, conferring it the above-mentioned relative tolerance as compared to CB. These cultivar differences were supported by Kumar et al. [57] studies, which demonstrated that mobilization of Fe at the root-rhizosphere interface seems to be achieved by either pH reduction or a drop in redox potential, or both, depending upon plant species and variety. Mehrotra et al. [58] reported that the ability to acquire and translocate iron in plants varies within the species and classifies plants as iron-efficient or iron-inefficient based on their response to Fe deficiency. Fe-efficient species or cultivars stimulate morphological, physiological, and biochemical changes, including root hair formation, proton extrusion, and Fe-CR activity, that make iron more available for root uptake. The tolerant cultivar, CN, thus belongs to this class. However, Fe-inefficient species are not capable of building up such mechanisms, or they build an inefficient one, so the sensitive cultivar, CB, belongs to this class. During their exposure to Fe limitation, other metabolic changes (at the physiological, biochemical, and molecular levels) are evident for the final adjustment of the plant response to Fe deficiency.

Finally, this study allowed us to identify some physiological and biochemical traits of the common bean response to iron deficiency, highlight the interdependence of the key metabolic functions, and identify useful indicators underlying the common bean tolerance and cultivar differences (Figure 10). All these metabolic traits operate coherently in CN, providing a robust defense system that gives it this relative tolerance. In CB, this system also exists, but it lacks efficiency. The identified traits of tolerance (FeUE, FeT, Fe-CRC, and RAC) are useful traits for further screening programs.

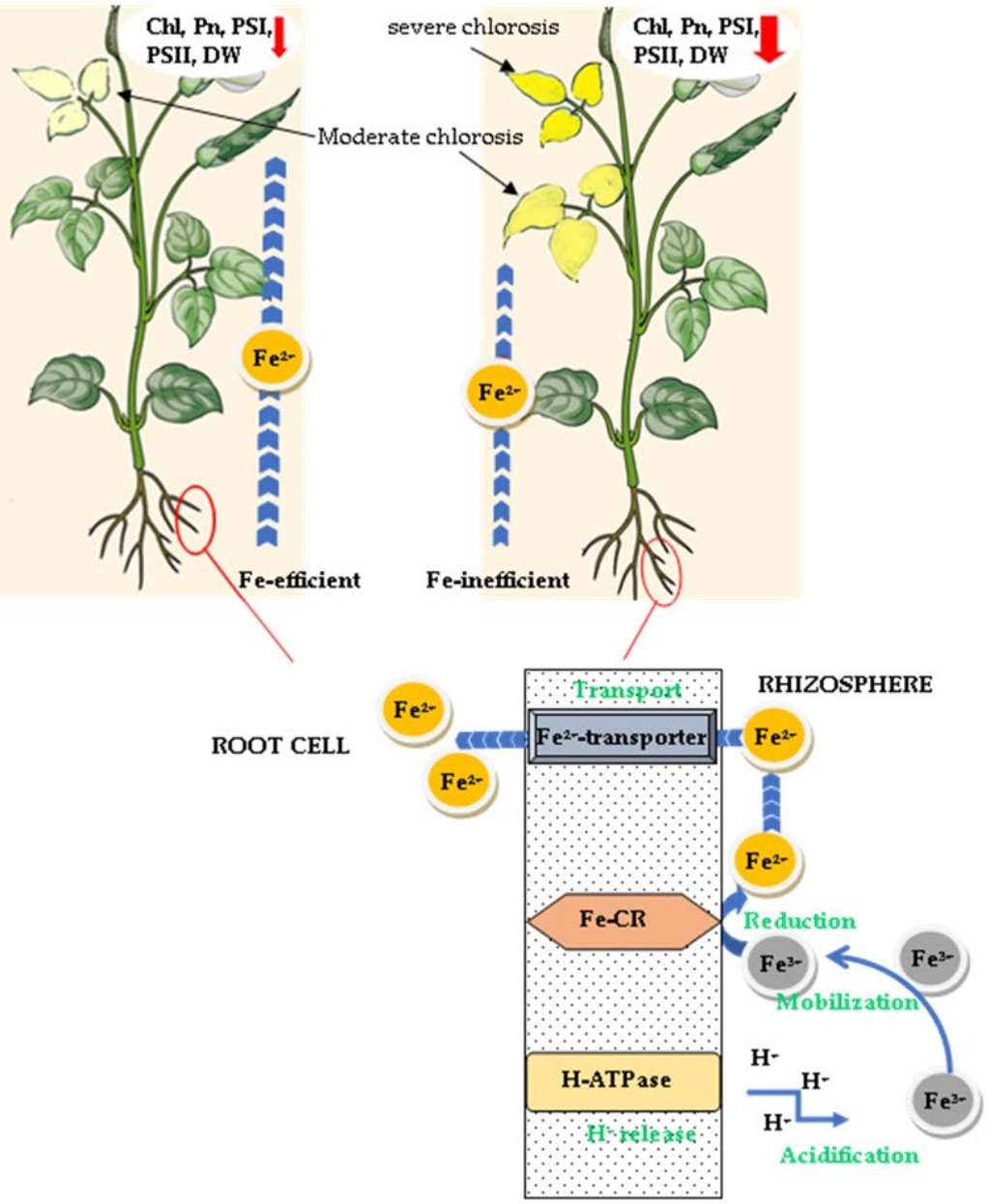

**Figure 10.** Schematic representation of the mechanisms underlying the cultivars differences in the response of common bean to iron deficiency. This mechanism operates effectively in *Fe-efficient* cultivar, CN, where the key metabolic reactions (Chl, Pn, PSI, PSII, DW) are less affected; and operates but with less effectivity in *Fe-inefficient* cultivar, CB, where these metabolic reactions are significantly affected. PM: plasma membrane.

**Author Contributions:** Formal analysis, K.N.; Data curation, K.N.; Writing—original draft, K.N.; Writing—review & editing, A.K.; Supervision, A.K.; Project administration, A.K.; Funding acquisition, A.K. All authors have read and agreed to the published version of the manuscript.

**Funding:** This study was supported by the Ministry of Higher Education and Scientific Research of Tunisia through the research Project DiVicia: Use and management of Vicia species for sustainability and resilience in biodiversity-based farming systems funded by PRIMA (Partnership for Research and Innovation in the Mediterranean Area).

**Data Availability Statement:** Not applicable.

**Conflicts of Interest:** The authors declare no conflict of interest.

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
