# Peer review of "The Key Physiological and Biochemical Traits Underlying Common Bean (Phaseolus vulgaris L.) Response to Iron Deficiency, and Related Interrelationships"

_agronomy, doi:10.3390/agronomy13082148_

Round 1

Reviewer 1 Report

The abstract is very well written, but may be improved by shortening. Minor orthographical editing is also needed. This research article can be substantially improved by rearranging the figures and the presentation of the results. In addition, shortening of the text may be also recommended, especially with regard to the discussion. 

The english language is fine, only minor editing is needed. 

Author Response

RESPONSE TO REVIEWER’S REPORTS

REVIEWER 1

Thank you very much for accepting to review our MS and present this report.

The MS was taken up again and subjected to a total revision of its scientific content, quality, style and presentation. All recommendations are taken in account, Figures and tables are readjusted to be clearer and more concise. All recommended amendments were added, some sentences were rephrased, and all changes were highlighted in green. Below our response to your report

  1. As recommended, the abstract was shortened.
  2. A number of sentences were rephrased,
  3. Numbers in chemical formulas were readjusted, superscript.
  4. Figure 1 (Picture of plant chlorosis) was removed as recommended by Reviewer 3.
  5. All figures are rearranged according to your recommendation and other reviewers recommendations.

- Figure 1 removed,

- Figure 2 data transferred to table 1,

- Figure 3 rearranged, new figure 1

- Figure 4 rearranged and quality improved, new figure 2

- Figures 5 and 6 rearranged and some mistakes removed, legends changed, comma in y axis changed by point … They are presented as one figure with the same x axis, new figure 3

- Figure 7 rearranged and presented as one panel, new figure 4

- Figure 8 readjusted, quality improved, comma in y equation changed by point… Because the presented data are different (different x, y axis), it was separated into 3 figures, new figure 5, figure 6, and figure 7

 - Same remarks for figure 9, new figure 8, figure 9

- Figure 10 was redesigned and clearly improved. It was designed by us.

  1. References

References were verified as recommended by the journal. They are adjusted accordingly.

  1. Discussion also improved as recommended. It was divided into 3 paragraphs as recommended by reviewer 3.

I will be in your service for all improvements you see.

Best regards

Reviewer 2 Report

The manuscript is very interesting and well written, even more authors proposed the mechanisms underlying the cultivars differences in the response of common bean to iron deficiency.

Only minor details should be solved to improve the quality of the article, they are inlcuded in the file as comments.

I suggest that a Principal Component Analysis should be perfomed in order to elucidate the realtionship among the studied variables to replace the correlations between two variables.

Author Response

RESPONSE TO REVIEWER’S REPORTS

REVIEWER 2

Thank you very much for accepting to review our MS and present this report.

The MS was taken up again and subjected to a total revision for its scientific content, quality, style and presentation. All recommendations are taken in account, Figures and tables are readjusted to be clearer and more concise. All recommended amendments and corrections were added, some sentences were rephrased, and all changes were highlighted in green. Below our response to your report

  1. Abstract as well as the overall text was revised for its grammatical presentation. Verb tense was adjusted in past,
  2. Keywords rearranged as recommended,
  3. Numbers in chemical formulas were readjusted, superscript,
  4. Formulas of calculation and analyzed parameters were rearranged,
  5. Figure 1 (Picture of plant chlorosis) was removed as recommended by Reviewer 3.
  6. All figures are rearranged according to all reviewers’ recommendations.

- Figure 1 removed,

- Figure 2 data transferred to table 1,

- Figure 3 rearranged, new figure 1

- Figure 4 rearranged and quality improved, new figure 2

- Figures 5 and 6 rearranged and some mistakes removed, legends changed, comma in y axis changed by point … They are presented as one figure with the same x axis, new figure 3

- Figure 7 rearranged and presented as one panel, new figure 4

- Figure 8 readjusted, quality improved, comma in y equation changed by point… Because the presented data are different (different x, y axis), it was separated into 3 figures, new figure 5, figure 6, and figure 7

 - Same remarks for figure 9, new figure 8, figure 9

- Figure 10 was redesigned and clearly improved

  1. Tables and tables legends were revised, rearranged and clearly improved according to the journal style
  2. For the recommended PCA for correlation

It is difficult to do it at this moment. Otherwise, correlations are not made between all measured and analyzed parameters. They are made according to a well-defined sequence in order to identify the common thread between precise parameters. photosynthesis, chlorophyll, and biomass with Fe; Fe with Fe-CR; Fe-CR with H-ATPase, and Pn with chlorophyll. They are not made between all parameters in an arbitrary way.

  1. 9. Discussion also improved as recommended. It was divided into 3 paragraphs as recommended by reviewer 3.

References were verified as recommended by the journal. They are adjusted accordingly.

I will be in your service for all improvements you see.

Best regards

Pr. Krouma Abdelmajid

Reviewer 3 Report

1.124,125 lines of chemical formula without subscript

2.Figure 1 Photo quality is not good, not clear, the picture also with watermark, if there is no way to replace a better picture, it is recommended to delete. State indicators generally include plant height, stem diameter, etc., 3.1 Title is not appropriate

3.The comments in the table should be written at the bottom of the table, not with the title, 268 rows in table 1 chlorophyll pigments, mg.g-1 MF does not need to appear in the table, spad index data with ', ' instead of '. ', need to change

4.Figure 2 b still uses ', ' instead of '. ', which needs to be changed, and the small problems in the text need to be strictly corrected

5.The data retention decimals in all tables are inconsistent and need to be modified. The table format is seriously inconsistent, and it needs to be uniformly arranged according to the requirements of the journal. If the journal has no special requirements, it should also be uniformly formatted.

6. There are two 3.3 in the title

7.Correlation analysis between the indicators in the Indicators and Physiological Indicators Recommendations can be expressed as a heat map, in which all indicators are analyzed for correlation, rather than each indicator in the map being analyzed separately.

8.Discussion should have a title, 4.1,4.2..., so that more clear

9.figure 10 map notes have problems, need to re-adjust the format

10.Full text typesetting, detail issues need attention

Author Response

RESPONSE TO REVIEWER’S REPORTS

REVIEWER 3

Thank you very much for accepting to review our MS and present this report.

The MS was taken up again and subjected to a total revision for its scientific content, quality, style and presentation. All recommendations are taken in account, Figures and tables are readjusted to be clearer and more concise. All recommended amendments and corrections were added, some sentences were rephrased, and all changes were highlighted in green. Below our response to your report

  1. Abstract as well as the overall text was revised for its grammatical presentation. Verb tense was adjusted in past,
  2. Numbers in chemical formulas were readjusted, superscript,
  3. Formulas of calculation and analyzed parameters were rearranged,
  4. Figure 1 (Picture of plant chlorosis) was removed,
  5. All figures are rearranged according to all reviewers’ recommendations.

- Figure 1 removed,

- Figure 2 data transferred to table 1,

- Figure 3 rearranged, new figure 1

- Figure 4 rearranged and quality improved, new figure 2

- Figures 5 and 6 rearranged and some mistakes removed, legends changed, comma in y axis changed by point … They are presented as one figure with the same x axis, new figure 3

- Figure 7 rearranged and presented as one panel, new figure 4

- Figure 8 readjusted, quality improved, comma in y equation changed by point… Because the presented data are different (different x, y axis), it was separated into 3 figures, new figure 5, figure 6, and figure 7

 - Same remarks for figure 9, new figure 8, figure 9

- Figure 10 was redesigned and clearly improved

  1. Tables and tables legends were revised, rearranged and clearly improved according to the journal style
  2. Data retention decimals in tables were verified and readjusted
  3. For correlation analyzis

It is difficult to do a PCA at this moment. Otherwise, correlations are not made between all measured and analyzed parameters. They are made according to a well-defined sequence in order to identify the common thread between precise parameters. photosynthesis, chlorophyll, and biomass with Fe; Fe with Fe-CR; Fe-CR with H-ATPase, and Pn with chlorophyll. They are not made between all parameters in an arbitrary way.

  1. 9. Discussion also improved as recommended. It was divided into 3 paragraphs as recommended.
  2. References were verified as recommended by the journal. They are adjusted accordingly.

I will be in your service for all improvements you see.

Best regards

Pr. Krouma Abdelmajid

Reviewer 4 Report

Dear Authors

The manuscript is interesting, and well written. However, minor revision is needed as attached.

The manuscript is int well written. Minor revision is needed to improve it.

Author Response

RESPONSE TO REVIEWER’S REPORTS

REVIEWER 4

Thank you very much for accepting to review our MS and present this report.

The MS was taken up again and subjected to a total revision for its scientific content, quality, style and presentation. All recommendations are taken in account, Figures and tables are readjusted to be clearer and more concise. All recommended amendments and corrections were added, some sentences were rephrased, and all changes were highlighted in green. Below our response to your report

  1. All corrections and amendments were made, verb tense in past, control instead of c, stressed instead of S, CB-C, CB-S, CN-S ….
  2. Numbers in chemical formulas were readjusted, superscript,
  3. Equation of calculation and analyzed parameters were rearranged and presentation improved,
  4. Figure 1 (Picture of plant chlorosis) was removed,
  5. All figures are rearranged according to all reviewers’ recommendations.

- Figure 1 removed,

- Figure 2 data transferred to table 1,

- Figure 3 rearranged, new figure 1

- Figure 4 rearranged and quality improved, new figure 2

- Figures 5 and 6 rearranged and some mistakes removed, legends changed, comma in y axis changed by point … They are presented as one figure with the same x axis, new figure 3

- Figure 7 rearranged and presented as one panel, new figure 4

- Figure 8 readjusted, quality improved, comma in y equation changed by point… Because the presented data are different (different x, y axis), it was separated into 3 figures, new figure 5, figure 6, and figure 7

 - Same remarks for figure 9, new figure 8, figure 9

- Figure 10 was redesigned and clearly improved

  1. Tables and tables legends were revised, rearranged and clearly improved according to the journal style, point instead of comma in some data …
  2. Data retention decimals in tables were verified and readjusted
  3. 9. Discussion also improved as recommended. It was divided into 3 paragraphs as recommended.
  4. References were verified as recommended by the journal. We used abbreviated journal name. Full name was used only for book series or proceedings

I will be in your service for all improvements you see.

Best regards

Pr. Krouma Abdelmajid
